# An Indoor Positioning Method Based on UWB and Visual Fusion

**DOI:** 10.3390/s22041394

**Published:** 2022-02-11

**Authors:** Pingping Peng, Chao Yu, Qihao Xia, Zhengqi Zheng, Kun Zhao, Wen Chen

**Affiliations:** 1Engineering Center of SHMEC for Space Information and GNSS, East China Normal University, Shanghai 200241, China; 51205904079@stu.ecnu.edu.cn (P.P.); 51205904027@stu.ecnu.edu.cn (Q.X.); zqzheng@ee.ecnu.edu.cn (Z.Z.); kzhao@ce.ecnu.edu.cn (K.Z.); wchen@sist.ecnu.edu.cn (W.C.); 2Shanghai Key Laboratory of Multidimensional Information Processing, East China Normal University, Shanghai 200241, China; 3Key Laboratory of Geographic Information Science, Ministry of Education, East China Normal University, Shanghai 200241, China

**Keywords:** indoor positioning, UWB, vision, fusion, PCA

## Abstract

Continuous positioning and tracking of multi-pedestrian targets is a common concern for large indoor space security, emergency evacuation, location services, and other application areas. Among the sensors used for positioning, the ultra-wide band (UWB) is a critical way to achieve high-precision indoor positioning. However, due to the existence of indoor Non-Line-of-Sight (NLOS) error, a single positioning system can no longer meet the requirement for positioning accuracy. This research aimed to design a high-precision and stable fusion positioning system which is based on the UWB and vision. The method uses the Hungarian algorithm to match the identity of the UWB and vision localization results, and, after successful matching, the fusion localization is performed by the federated Kalman filtering algorithm. In addition, due to the presence of colored noise in indoor positioning data, this paper also proposes a Kalman filtering algorithm based on principal component analysis (PCA). The advantage of this new filtering algorithm is that it does not have to establish the dynamics model of the distribution hypothesis and requires less calculation. The PCA algorithm is firstly used to minimize the correlation of the observables, thus providing a more reasonable Kalman gain by energy estimation and the denoised data, which are substituted into Kalman prediction equations. Experimental results show that the average accuracy of the UWB and visual fusion method is 25.3% higher than that of the UWB. The proposed method can effectively suppress the influence of NLOS error on the positioning accuracy because of the high stability and continuity of visual positioning. Furthermore, compared with the traditional Kalman filtering, the mean square error of the new filtering algorithm is reduced by 31.8%. After using the PCA-Kalman filtering, the colored noise is reduced and the Kalman gain becomes more reasonable, facilitating accurate estimation of the state by the filter.

## 1. Introduction

In recent years, with the rapid development of location-based service (LBS) applications, indoor positioning research in large indoor spaces (such as underground parking lots and large commercial complexes) has received increasing attention from scholars. High-accuracy indoor positioning technology has become a common basis for public safety, intelligent management, business analysis, LBS, and other fields [1]. At present, many methods are used to build indoor positioning systems, such as the time-based positioning method [2,3,4], radio fingerprints [5,6,7,8], and dead reckoning [9,10]. Among these methods, the UWB positioning system supports very accurate timestamps and direct path selection capabilities based on pulse radio, with nanoscale time resolution. It is a critical way to achieve high-precision indoor positioning. In the time-based ranging, the high-resolution timestamp is used to measure the propagation time accurately, and the direct path selection is used to accurately classify the shortest signal path corresponding to the UWB radio distance in a multipath environment. Currently, the position accuracy can usually be improved by increasing the number of base stations, thus improving clock synchronization accuracy and data noise reduction. However, because the acoustic waves are easily affected by the environment in the process of propagation, the stability and positioning accuracy are not yet ideal and the usage scenarios of a positioning system consisting of single means is very limited. Therefore, the research and development of accurate and reliable indoor positioning methods can not only improve the accuracy and reliability of indoor target positioning, but also have the potential for broad market application.

Different solutions to the above challenges have been proposed. A fusion framework was designed in [11], which is used to calibrate the inertial sensor when the ultra-wideband sensor is available, and the system switches to the inertial sensor’s solution for output when the ultra-wideband signal is weak. However, although the inertial sensors are able to provide highly accurate positioning data in a short period of time, there is a large cumulative error due to device drift. A previous study [12] proposed three algorithms based on the recursive Bayesian methods to fuse inertial and ranging information to improve the localization accuracy. However, the scheme has some limitations due to the error accumulation of inertial guidance and the time of arrival (TOA) method requires strict clock synchronization between base stations. In [13], a systematic approach was presented for the autonomous exploration and mapping of an unknown environment using dual RGB-D sensors. Although the developed system and method have high robustness and efficiency, the large amount of computation due to the richness of information contained in the 3D point cloud map makes the positioning system insufficient in real time.

This contribution proposes a novel indoor positioning method fusing the UWB and vision because the UWB and visual technology are highly complementary in the positioning process. The UWB positioning systems require an infrastructure of three or more anchor points, namely, the base station (BS) and Line-of-Sight (LOS) channels, without any barriers between the positioning tag and the anchor point. This is the limitation of the indoor positioning system based on the UWB. To provide continuous and stable positioning services in the NLOS environment or in the absence of a certain number of base stations, visual positioning systems are used to assist UWB systems in indoor positioning. Unlike UWB’s active positioning, the visual positioning system can detect the movement of the target pedestrian using indoor fixed cameras, without the need for the target to wear additional equipment. It is a passive positioning method with low cost and high scalability [14]. In recent years, computer vision relying on machine learning has been developed rapidly and widely used. An increasing number of industries are using computer vision for self-innovation, face recognition, augmented reality, video analysis, etc., which are currently popular research topics. As a form of traditional indoor positioning technology that has been developed for many years, it is necessary to expand new ideas and seek new changes in order to develop and progress, and computer vision is a good auxiliary technology to promote the development of indoor positioning.

In this method, pedestrian images are first sent into the network model of the YOLOv3 algorithm to detect pedestrians. Then, pedestrians are tracked by the DeepSort algorithm and each target is assigned a tracker ID. We next calculate the world coordinates of the target through the coordinate transformation model. At this point, the initial positioning coordinates of the vision can be matched with the initial positioning coordinates of the UWB. After matching successfully, the identity of the target of visual positioning is known because the UWB tag can provide the identity of the pedestrian, and the real coordinates of the pedestrian can be obtained by the combination of the visual positioning and UWB positioning. Due to the continuity of visual positioning, this method can reduce the positioning error of the UWB, and the positioning results of the UWB can also provide positioning information for blind areas of the visual positioning. Therefore, the probability of not detecting pedestrians is reduced.

At the same time, in an indoor positioning system, the robustness of the filtering algorithm is also crucial to accuracy. The Kalman filtering algorithm is widely used in navigation and positioning [15,16,17]. However, in indoor locations, the continuous dynamic process and slow clock change lead to time correlation between noise calendar elements; that is, the noise is colored [18]. Since there is a large quantity of colored noise in indoor environments and the Kalman filter only performs well in the environment of white noise [19], the new unscented Kalman filter [20] was presented for nonlinear systems with colored measurement noise based on the augmented measurement information method. However, as the dimension of the augmented system increases, the order of the filter also increases, resulting in a larger calculation. In this paper, we propose a Kalman filter algorithm based on principal component analysis (PCA). Firstly, the PCA algorithm is used to estimate the noise and signal energy of the measured dataset, to obtain a more reasonable Kalman gain. Then, the data denoised by PCA is substituted into the Kalman prediction equation, which further improves the performance of the filter.

## 2. An Overview of Indoor Positioning Methods of the UWB

Received signal strength (RSS), angle of arrival (AOA), and time of arrival or time difference of arrival (TOA/TDOA) are classical UWB positioning methods. At present, the TOA and TDOA methods are most commonly used. However, TDOA does not require clock synchronization between base stations and tags, and only requires time consistency among base stations. Therefore, the TDOA method is adopted in this paper.

### 2.1. Description of the TDOA Positioning Method

The positioning principle of TDOA is shown in Figure 1, and the coordinates of the four base stations in the figure below are $BS_{i}(X_{i},Y_{i}),(i=1,2,3,4)$, where $BS_{1}$ is set as the master base station, and $BS_{i}(X_{i},Y_{i}),(i=2,3,4)$ are slave stations. When a pedestrian moves around a base station with a UWB tag, the tag emits UWB signals periodically. Suppose that moments when $BS_{i}(i=1,2,3,4)$ receive UWB signals are $t_{i}$, respectively, then the TDOA equation can be obtained as Equation (1).
(1)$$R_{i,1}=c(t_{i}-t_{1})=\sqrt{{\left(x_{i}-a\right)}^{2}+{\left(y_{i}-b\right)}^{2}}-\sqrt{{\left(x_{1}-a\right)}^{2}+{\left(y_{1}-b\right)}^{2}}$$
where $R_{i,1}$ is represented as the single difference between the duration of the tag to the base station i and that of the tag to the master station, *c* is the velocity of light, and (a,b) is the coordinate of the pedestrian, namely, the tag. Equation (2) can be easily obtained:(2)$$\sqrt{{\left(x_{i}-a\right)}^{2}+{\left(y_{i}-b\right)}^{2}}-\sqrt{{\left(x_{1}-a\right)}^{2}+{\left(y_{1}-b\right)}^{2}}=R_{i,1},i=2,3,4$$

Obviously, the problem of solving the tag coordinate is transformed into deducing the optimum solution of Equation (2).

### 2.2. Chan-Taylor Cascade Location Algorithm

The Taylor algorithm is a recursive algorithm with strong robustness and high precision. In each recursion, the estimated position of the tag is updated by solving the local least square method of the TDOA measurement error. However, the algorithm needs the value close to the real tag position coordinate as the initial value, otherwise the Taylor algorithm may not converge, and it is difficult to locate the tag. Therefore, in order to strengthen the convergence of the Taylor algorithm, the Chan algorithm is adopted in this paper to preprocess the measured data. The estimated value is taken as the expansion point of the Taylor algorithm.

In combination with Equations (1) and (2), we can obtain:(3)$$ax_{i,1}+by_{i,1}+R_{i,1}R_{1}=\frac{1}{2}(K_{i}-K_{1}-R_{i,1}^{2})$$
where $K_{i}=x_{i}^{2}+y_{i}^{2}$, $x_{i,1}=x_{i}-x_{1}$, $y_{i,1}=y_{i}-y_{1}$, (i=2,3,4), taking $x,y,R_{1}$ as independent variables. Equation (3) is linearized as follows:(4)$$Gz_{a}=h$$
where $z_{a}={\left[\begin{array}{ccc} a & b & R_{1} \end{array}\right]}^{T}$, $h=-\frac{1}{2}\left[\begin{array}{c} R_{2,1}^{2}-K_{2}+K_{1}\\ R_{3,1}^{2}-K_{3}+K_{1}\\ R_{4,1}^{2}-K_{4}+K_{1} \end{array}\right]$, $G=-\left[\begin{array}{ccc} x_{2,1} & y_{2,1} & R_{2,1}\\ x_{3,1} & y_{3,1} & R_{3,1}\\ x_{4,1} & y_{4,1} & R_{4,1} \end{array}\right]$.

The first-time estimation is obtained by using the least square method:(5)$$z_{a}={\left[\begin{array}{ccc} a & b & R_{1} \end{array}\right]}^{T}={\left(G^{T}Q^{-1}G\right)}^{-1}G^{T}Q^{-1}h$$
where *Q* represents the standard deviation of TDOA measurement error.

According to Equation (5), the coordinates of the first solution of the tag can be obtained, but they are only fuzzy estimates. In fact, the variable $R_{1}$ in $z_{a}$ is related to the position P(a,b) of the tag. In order to reduce the error caused by the correlation of variables in $z_{a}$, variables in $z_{a}$ are taken as stochastic. If the linear relationship between the squared variables is known, a new set of error equations is constructed using the constraint conditions such as the estimated position obtained by Equation (5) and additional variable $R_{1}$, and then the second-time least square estimation is performed. The expression is given as:(6)$$z_{a1}=\left[\begin{array}{c} {\left(a-x_{1}\right)}^{2}\\ {\left(b-y_{1}\right)}^{2} \end{array}\right]={\left(G^{\prime }{Q^{\prime }}^{-1}G^{\prime }\right)}^{-1}{G^{\prime }}^{T}{Q^{\prime }}^{-1}h^{\prime }$$
where $G^{\prime }=\left[\begin{array}{cc} 1 & 0\\ 0 & 1\\ 1 & 1 \end{array}\right]$, $h^{\prime }=\left[\begin{array}{c} {\left(z_{a}(1)-x_{1}\right)}^{2}\\ {\left(z_{a}(2)-y_{1}\right)}^{2}\\ {\left(z_{a}(3)\right)}^{2} \end{array}\right]$, $Q^{\prime }=4B\mathrm{cov}(z_{a})B$, $B=diag(z_{a}(1)-x_{1},z_{a}(2)-y_{1},R_{1})$.

The estimated coordinates of the tag are:(7)$$(\bar{a},\bar{b}{)}^{T}=\pm \sqrt{z_{a1}}+{\left(x_{1},y_{1}\right)}^{T}$$

Then, the estimated value obtained by the Chan algorithm is substituted into the Taylor algorithm, and the following equation is defined as:(8)$$f(a,b)=R_{i}-R_{1}=\sqrt{{\left(x_{i}-a\right)}^{2}+{\left(y_{i}-b\right)}^{2}}-\sqrt{{\left(x_{1}-a\right)}^{2}+{\left(y_{1}-b\right)}^{2}}$$

Assuming that Z=f(a,b) has the derivative of order n+1 in a neighborhood of point $\left(a_{0},b_{0}\right)$, the Taylor series expansion of the equation on a point $\left(a_{0}+\alpha ,b_{0}+\beta \right)$ in the neighborhood is:(9)$$f(a_{0}+\alpha ,b_{0}+\beta )=f(a_{0},b_{0})+\alpha \frac{\partial f(a_{0},b_{0})}{a}+\beta \frac{\partial f(a_{0},b_{0})}{b}+\ldots$$

The initial value $\left(\bar{a},\bar{b}\right)$ is substituted into Equation (9), and the Taylor series expansion is performed at this coordinate point, which is simplified as follows:(10)$$h_{1}=G_{1}\Delta +\Psi$$
where $G_{1}=\left[\begin{array}{cc} (x_{1}-\bar{a}/R_{1}-(x_{2}-\bar{a})/R_{2} & (y_{1}-\bar{b})/R_{1}-(y_{2}-\bar{b})/R_{2}\\ (x_{1}-\bar{a})/R_{1}-(x_{3}-\bar{a})/R_{3} & (y_{1}-\bar{b})/R_{1}-(y_{3}-\bar{b})/R_{3}\\ (x_{1}-\bar{a})/R_{1}-(x_{4}-\bar{a})/R_{4} & (y_{1}-\bar{b})/R_{1}-(y_{4}-\bar{b})/R_{4} \end{array}\right]$, $\Delta =\left[\begin{array}{c} \Delta a\\ \Delta b \end{array}\right]$.

If the weighted least square method is adopted to process the above equation, the Taylor iterative increment Δ can be obtained:(11)$$\Delta ={\left(G_{1}{}^{T}Q^{-1}G_{1}\right)}^{-1}G_{1}{}^{T}Q^{-1}h_{1}$$
where Q is the covariance matrix of the TDOA observations of the system. The increment Δ obtained in each iteration is added back to the coordinates of the previous time, and the coordinates iterate until the Δ is less than the preset threshold. The output result is the final coordinate of the tag.

### 2.3. Robust Kalman Filtering Algorithm Based on the PCA

White noise is a more ideal noise, with no correlation between adjacent ephemeris elements and zero covariance. Colored noise is a kind of noise commonly encountered in practical engineering systems, where the noise at each moment in the noise sequence is correlated with the noise at another moment, and the covariance is not zero. The Kalman filter algorithm itself is based on the proposed white noise, and it uses the white noise covariance involved in the state update process. By comparison, colored noise produce perturbations in the state update, thus reducing the accuracy of the Kalman filter algorithm. However, since most of the actual engineering is colored noise, the filter’s effectiveness, and stability is reduced. Moreover, the wrong estimation of measurement noise will affect the Kalman gain. Therefore, this paper proposes a Kalman filter algorithm based on PCA. The principle of the PCA method is that a set of correlated data is orthogonally transformed to construct orthogonal eigenvectors, and with these orthogonal eigenvectors as the basis, the corresponding coefficients constitute the principal components. Arranged in descending order of the energy contributed by the principal components, only the first few principal components are needed to encompass the majority of the energy in the original data, so an important application of the PCA analysis method is to eliminate noise.

Firstly, the PCA algorithm is used to transform the original dataset to minimize the cross-correlation of data samples. Then, the dimension with high energy is regarded as the primary dimension and the data of this dimension is reserved as the useful signal. The other dimension is the noise irrelevant to the useful signal, which is the second dimension. Thus, the energy of the useful signal and the noise signal can be obtained, and their ratio can be used to measure the reliability of the dataset, which provides a partial basis for the value of Kalman gain. At the same time, all the values of the secondary dimensions are set to zero, that is, all noise signals irrelevant to the useful signal are removed, and the measurement dataset is updated. The de-noised data can be obtained by re-mapping the dataset to the original coordinate axis, which is used as the measurement value of the Kalman filter, and the positioning result is output after filtering.

We assume that $S_{k}=[\begin{array}{ccc} x_{1} & \ldots & x_{m} \end{array}]$ is the original positioning data calculated at moment k, and $I_{i,j}=\{S_{i},\ldots ,S_{j}\}$ represents the original positioning dataset that needs PCA processing from moment i to j. Suppose that the input sample is the measured data from moment 1 to moment n, and the sample number is greater than the number of characteristic parameters, and the matrix M formed is as follows:(12)$$M=\left[\begin{array}{c} S_{1}\\ \vdots \\ S_{n} \end{array}\right]=\left[\begin{array}{ccc} x_{11} & \cdots & x_{1m}\\ \vdots & \ddots & \vdots \\ x_{n1} & \ldots & x_{nm} \end{array}\right]$$
where *m* and *n* are number of samples and characteristic parameters, respectively. $M_{j}$ can be normalized as:(13)$$A_{j}=\frac{M_{j}-\mu_{j}}{\sqrt{\sigma_{j}^{2}}}\;j=1,2,\ldots ,m$$
where $\mu_{j}=E\left(M_{j}\right)$, $\sigma_{j}^{2}=D(M_{j})$. *E* and *D* denote the expectation and variance of $M_{j}$, respectively.

The variance of each column of matrix A is one, and the expectation is zero. The correlation coefficient matrix R of A can be written as:(14)$$R=\left[\begin{array}{ccc} r_{11} & \cdots & r_{1m}\\ \vdots & \ddots & \vdots \\ r_{m1} & \ldots & r_{mm} \end{array}\right]$$

According to Equation (14), the eigenvalue λ of correlation coefficient matrix R and its corresponding eigenvectors are calculated by:(15)|λI−R|=0

Equation (16) is used to calculate the variance contribution rate of each eigenvector, and the eigenvector whose variance contribution rate reaches more than 85% is obtained:(16)$$\sum_{k=1}^{l}P_{count}(k)=\frac{\sum_{k=1}^{l}\lambda_{k}}{\sum_{i=1}^{m}\lambda_{i}}$$
where $P_{count}(k)$ is the contribution rate of the principal element *k*, and l is the number of principal elements.

In this paper, input information is divided into two parts: the useful signal and the noise signal. It is assumed that the colored noise is independent of the useful signal, and the energy of the useful signal is greater than that of the noise signal. Then the original data will obtain two eigenvalues, $\lambda_{1}$ and $\lambda_{2}$, after PCA processing, and the corresponding eigenvectors are $\beta_{1}$ and $\beta_{2}$, respectively. If $\lambda_{1}>\lambda_{2}$, and $E=\frac{\lambda_{1}}{\lambda_{2}+\lambda_{1}}$ is represented as the energy ratio to the useful signal and input signal, the degradation factor α is:(17)$$\alpha =\left\{ \begin{array}{l} \begin{array}{cc} 1, & E>0.95 \end{array}\\ \frac{1}{E},0.85\leq E\leq 0.95\\ {\left(E\right)}^{-2},0<E<0.85 \end{array}\right.$$

The original data are mapped to the coordinate frame that is composed of eigenvectors, and values under the primary axis are obtained. At the same time, the secondary coordinate values are set to zero. Then, we take the inverse of the square matrix of all the eigenvectors, multiply the denoised data, and map it to the original coordinates. The denoised datasets preprocessed by PCA are acquired, set to ${\tilde{I}}_{1,n}=\{X_{1},\ldots ,X_{n}\}$, where $X_{k}=[\begin{array}{ccc} {\tilde{x}}_{1} & \ldots & {\tilde{x}}_{m} \end{array}]$, k=1,2,…,n, and $X_{k}$ is the observation vector of $S_{k}$ after denoising. Finally, the denoised data are substituted into the state and the observation equations:(18)$$X_{k}=FX_{k-1}+\omega_{k}$$
(19)$$Z_{k}=H_{k}X_{k}+\nu_{k}$$
where $Z_{k}$ is the observation vector, F is the state transition matrix, $\omega_{k}$ and $\nu_{k}$ are mutually independent and normally distributed white noise, and their statistical characteristics are as follows:(20)$$\begin{array}{l} E(\omega_{k})=0,\mathrm{cov}(\omega_{k},\omega_{j})=Q_{k}\delta_{kj}\\ E(\nu_{k})=0,\mathrm{cov}(\nu_{k},\nu_{j})=R_{k}\delta_{kj}\\ \mathrm{cov}(\omega_{k},\nu_{k})=0,\delta_{kj}=\left\{ \begin{array}{l} \begin{array}{cc} 1 & k=j \end{array}\\ \begin{array}{cc} 0 & k\neq j \end{array} \end{array}\right. \end{array}$$

In Equation (20), $R_{k}$ and $Q_{k}$ are the covariance matrices of $\nu_{k}$ and $\omega_{k}$, respectively, and are independent each other.

In the data update stage, the filtering algorithm proposed in this paper introduces the robust factor matrix $\phi =diag\left[\begin{array}{cc} \alpha & \alpha \end{array}\right]$, so the observation error covariance matrix is ${\bar{R}}_{k}=\phi R_{k}$, and the measurement update process from moment k−1 to moment k is as follows:(21)$$\left\{ \begin{array}{l} {\bar{K}}_{k}=P_{k,k-1}H_{k}{}^{T}{\left[H_{k}P_{k,k-1}H_{k}{}^{T}+{\bar{R}}_{k}\right]}^{-1}\\ {\hat{X}}_{k}={\hat{X}}_{k,k-1}+{\bar{K}}_{k}\left[Z_{k}-H_{k}{\hat{X}}_{k,k-1}\right]\\ P_{k}=\left[I-{\bar{K}}_{k}H_{k}\right]P_{k,k-1} \end{array}\right.$$

In Equation (21), ${\hat{X}}_{k,k-1}=F{\hat{X}}_{k-1}$, $P_{k,k-1}=FP_{k-1}F^{T}+Q_{k-1}$, and the degradation factor is used to update and modify the observation covariance matrix of system noise, which is put into the gain matrix. Furthermore, the input measurement data can effectively filter out the colored noise in the positioning process, thus reducing the state estimation error and improving the stability of the system filtering algorithm. The PCA–Kalman algorithm block diagram is represented in Figure 2.

## 3. The Mechanism of Visual Target Tracking and Positioning

### 3.1. Multi-Target Tracking Algorithm

This paper uses target detection and tracking algorithms for visual positioning, respectively, YOLOv3 and DeepSort. During the matching process, the target image is divided into S×S cells by the target detection algorithm, and three anchor frames detect each cell for the target whose center point falls on this cell; then, the tensor with the size S×S×3×(4+1+C) is output. It contains the central coordinate, height and width of the prediction box, confidence degree, and C class probabilities. Finally, the non-maximum suppression (NMS) method is used to determine the coordinate of the detection target and the predicted value of the category. Confidence is defined as follows:(22)$$confidence=P_{r}\left(object\right)*IOU_{perd}^{truth}$$
where confidence is the confidence of the bounding box, $P_{r}\left(object\right)$ is the category probability of a certain target in the network, $IOU_{perd}^{truth}$ is the intersection over union ratio between the bounding box and the real box.

After performing target detection on one frame of the image using the YOLOv3 algorithm, the DeepSort algorithm assigns a unique ID number to the pedestrian in the image. It tracks the pedestrian in the next frame. The algorithm uses the Kalman filtering to combine the predicted value and measured value to make an optimal estimation of the system state. Based on the Hungarian matching algorithm, a cascade matching method is used to prioritize the frequently updated tracker. At the same time, the DeepSort algorithm introduces the appearance feature based on the classical SORT algorithm because, in practical application, simple motion information is not enough to accurately achieve multi-target tracking. To improve the prediction accuracy of the target in the case of variable motion and turning, the DeepSort algorithm first stores the feature vectors of the last several frames successfully associated with each tracking target. It then calculates the similarity between the appearance features of the detected object in each current frame and the appearance features stored previously. Therefore, the DeepSort algorithm has the feature of deep correlation, which makes the tracking accuracy higher than that of the SORT algorithm, and the ID jump is reduced effectively. Thus, it is more applicable to indoor positioning scenes.

Figure 3 shows the detection and tracking results based on visual target tracking. When two pedestrian targets appear in the picture, the targets will be framed with a rectangular frame. The ID serial numbers assigned to the target tracker will be displayed above the rectangular frame, as shown with person-1 and person-2 in the figure. In the subsequent screen, the DeepSort algorithm will determine whether the target appearing in the current frame is the same person as the target in the previous frame to achieve continuous tracking, as shown in Figure 4. In the subsequent positioning, the tracker IDs of the two people in the picture are always unchanged, and the plantar pixel coordinates are marked in the picture in Figure 4 to visualize the tracks of the two targets.

### 3.2. Principle of Coordinate Frame Transformation

The result of the pedestrian detection of the visual image is the pixel coordinate, and the calculation formula for transforming the pixel coordinate (u,v) into the image coordinate (x,y) is as follows:(23)$$\left\{ \begin{array}{l} u=\frac{x}{dx}+u_{0}\\ v=\frac{y}{dy}+v_{0} \end{array}\right.$$

In Equation (23), dx,dy represents the physical size of each pixel along the coordinate axis, in mm/pixel, and $\left(u_{0},v_{0}\right)$ represents the translation of the origin of the two-dimensional plane coordinate system.

By transforming Equation (23) into matrix form, we can obtain:(24)$$\left[\begin{array}{c} u\\ v\\ 1 \end{array}\right]=\left[\begin{array}{ccc} \frac{1}{dx} & 0 & u_{0}\\ 0 & \frac{1}{dy} & v_{0}\\ 0 & 0 & 1 \end{array}\right]\left[\begin{array}{c} x\\ y\\ 1 \end{array}\right]$$

Conversion of the image coordinate (x,y) to the camera coordinate is undertaken using the perspective projection transformation. On the assumption that focal distance f is given, conversion can be carried out by the following formula:(25)$$z_{c}\left[\begin{array}{c} x\\ y\\ 1 \end{array}\right]=\left[\begin{array}{cccc} f & 0 & 0 & 0\\ 0 & f & 0 & 0\\ 0 & 0 & 1 & 0 \end{array}\right]\left[\begin{array}{c} x_{c}\\ y_{c}\\ z_{c}\\ 1 \end{array}\right]$$
where $\left(x_{c},y_{c},z_{c}\right)$ represents the three-dimensional coordinate in the camera coordinate system.

Converting camera coordinates to world coordinates involves rotation and translation of coordinates, rotation of different angles around different coordinate axes, and calculation of world coordinates $\left(x_{w},y_{w},z_{w}\right)$ using the following formula:(26)$$\left[\begin{array}{c} x_{c}\\ y_{c}\\ z_{c} \end{array}\right]=R\left[\begin{array}{c} x_{w}\\ y_{w}\\ z_{w} \end{array}\right]+T$$
where *R* is the rotation matrix and *T* is the offset vector.

## 4. Positioning Model Based on the UWB and Visual Fusion

Because of NLOS error and unstable signal transmission in the UWB positioning, this paper proposes an indoor positioning method that combines the vision and UWB. During positioning, the pedestrian holds a UWB tag, which provides a unique ID number. In addition, a monocular camera should be installed above the indoor positioning area. The camera captures the image in real time and sends the image into the network model of the YOLOv3 algorithm to detect the pedestrian, then tracks the pedestrian in the picture using the DeepSort algorithm. In the case of image localization only, once the pedestrian target is detected, the target is fixed with a rectangular frame. Then, the plantar pixel coordinates are transformed into world coordinates. These coordinates are matched with the result of UWB location only by Euclidean distance. After matching, the federated Kalman filter is applied to the image and UWB location results to obtain the real position of the pedestrian. The flowchart of the fusion model is shown in Figure 5.

### 4.1. Pedestrian Identity Matching Process

We calculate the Euclidean distance of the UWB positioning coordinates and all visual positioning coordinates one by one, and the calculation formula is as follows:(27)$$d(X,Y)=\sqrt{{\left(x_{1}-y_{1}\right)}^{2}+{\left(x_{1}-y_{1}\right)}^{2}+\ldots +{\left(x_{m}-y_{m}\right)}^{2}}=\sqrt{\sum_{i=1}^{m}{\left(x_{i}-y_{i}\right)}^{2}}$$
where $X=(x_{1},\ldots ,x_{m}),Y=(y_{1},\ldots ,y_{m})$ represent two points in the m-dimensional space.

In this paper, n vision and UWB anchor points need to be matched respectively, and $d_{ij}$ is the Euclidean distance between the vision positioning coordinate *i* and the UWB positioning coordinate *j*. $k_{ij}$ is the decision variable. When $k_{ij}$ = 1, it means that the location *i* of vision is matched with the location *j* of UWB; otherwise, $k_{ij}$ = 0. In order to ensure all points match successfully and result in the shortest total Euclidean distance, the mathematical model of the matching problem can be expressed as:(28)$$\min f=\sum_{i}\sum_{j}d_{ij}k_{ij}$$

In Equation (28), $\sum_{i}k_{ij}=1,j=1,2,\ldots ,n$, $\sum_{j}k_{ij}=1,i=1,2,\ldots ,n$, $k_{ij}$ = 1 or 0 are the constraint conditions of the above equation. Therefore, this task can be transformed into a balanced assignment problem, which can be solved quickly by the Hungarian algorithm. Aiming at the special form of the standard assignment model (square matrix) and based on the independent zero element theorem proposed by Hungarian mathematician Konig, the algorithm can obtain the optimal solution of the model using only matrix transformation [21,22].

### 4.2. The UWB and Visual Fusion Positioning Algorithm Based on the Federated Kalman Filter

After successful pedestrian identity matching, continuous high-precision pedestrian positioning can be achieved through the UWB and visual fusion positioning. As mentioned earlier, the visual positioning method has higher continuity and stability, so the advantage of this article’s fusion method lies in using the visual positioning stability to adjust the gross error in the process of UWB positioning and NLOS error. When the UWB signal is blocked or the tag cannot launch the signal, the visual positioning results can be used for supplementary positioning. Similarly, when the visual target detection result is lost, the target enters the visual blind area, and the visual detection is missed, the UWB data are switched to supplement the visual positioning result until the visual positioning succeeds again.

When both the visual and UWB location have positioning results, the federated Kalman filter algorithm combines the two positioning results to improve the pedestrian location accuracy [23].

We assume that the estimation result of the visual positioning state is ${\hat{X}}_{1}=[\begin{array}{ccc} a_{1} & \ldots & a_{m} \end{array}]$ and that of the UWB positioning state is ${\hat{X}}_{2}=[\begin{array}{ccc} b_{1} & \ldots & b_{m} \end{array}]$, and the corresponding estimation error variances are $P_{11}$ and $P_{22}$. Consider that the global state estimation ${\hat{X}}_{g}$ after fusion is a linear combination of local state estimation, namely:(29)$${\hat{X}}_{g}=W_{1}{\hat{X}}_{1}+W_{2}{\hat{X}}_{2}$$
where $W_{1}$ and $W_{2}$ are undetermined weighted matrices.

The global state estimation ${\hat{X}}_{g}$ should satisfy the following two conditions:

(1)If the local state estimation ${\hat{X}}_{1}$ and ${\hat{X}}_{2}$ are unbiased estimates, then ${\hat{X}}_{g}$ should also be an unbiased estimate, namely:(30)$$E\left[X-{\hat{X}}_{g}\right]=0$$
where X is the actual state.(2)The estimated state error covariance matrix of ${\hat{X}}_{g}$ is the smallest, that is
(31)$$P_{g}=E\left[\left[X-{\hat{X}}_{g}\right]{\left[X-{\hat{X}}_{g}\right]}^{T}\right]=min$$

Combining with condition (1), we can obtain:(32)$$P_{g}=P_{11}-W_{2}{\left[P_{11}-P_{12}\right]}^{T}-\left[P_{11}-P_{12}\right]W_{2}^{T}+W_{2}\left[P_{11}-P_{12}-P_{21}+P_{22}\right]W_{2}^{T}$$

In order to minimize $P_{g}$, we can calculate $W_{2}$ as follows:(33)$$W_{2}=\left[P_{11}-P_{12}\right]{\left[P_{11}-P_{12}-P_{21}+P_{22}\right]}^{-1}$$

At the same time, since ${\hat{X}}_{1}$ and ${\hat{X}}_{2}$ are unrelated, we can get:(34)$${\hat{X}}_{g}={\left[P_{11}^{-1}+P_{11}^{-1}\right]}^{-1}\left(P_{11}^{-1}{\hat{X}}_{1}+P_{22}^{-1}{\hat{X}}_{2}\right)$$

It can be proved that the global optimal estimation is better than the local estimation, that is $P_{g}<P_{11},P_{g}<P_{22}$ [24].

We provide a concrete example to better understand the process and usefulness of the federated Kalman filter algorithm. In the experiments of this paper, the states we estimate are two-dimensional coordinate values. Therefore, the algorithmic process of localization fusion after pedestrian identity matching is specified as follows:
Step 1:Input the visual and UWB localization values of the pedestrian, i.e., two-dimensional coordinate values.Step 2:Initialize the error variance *P*_11_ and *P*_22_.Step 3:Calculate the global optimal state estimate according to Equation (34).Step 4:Output results of the fusion positioning.


## 5. Experiment Results and Analysis

The test environment was the corridor on the fourth floor of the Information Building of East China Normal University. The experimental environment is shown in Figure 6. Four UWB base stations were arranged along the direction of the red arrow in the figure. Framed by the green rectangular frame is the column on the corridor, and behind the column is the blind spot of the camera. Each positioning base station is connected to the workstation through a switch and the camera transmits the collected video stream to the workstation for processing.

### 5.1. Experimental Results of the UWB and Visual Fusion Positioning

We selected two different experimental routes to verify the accuracy of the fusion positioning algorithm. The experimenters denoted person-1 and person-2 started from different starting points of the asterisk in Figure 7. They walked along different paths on the ground simultaneously, holding the UWB tags to obtain the UWB positioning data in the process of pedestrian movement. Furthermore, to make the experiment scene general, the route of the experimenter includes the visual blind area behind the column, as shown in Figure 8.

In the experiment, both the UWB and visual methods were used to detect two pedestrian targets, and the Chan–Taylor cascade algorithm was used to determine the UWB data. The positioning scatter results and actual track are shown in Figure 9a. Simultaneously, the pixel coordinates of pedestrians in the continuous image frames are calculated, and then the world coordinates are converted. The positioning results and the real track are shown in Figure 9b. When there are only the visual location results or the UWB location results in a certain period, the Kalman filtering obtains the pedestrian location results. When the positioning results of the two means are available at a certain time, the fusion positioning algorithm provides the pedestrian positioning results. The fusion positioning results and the real trajectory are shown in Figure 9c. In Figure 9, blue is the track of experimenter person-1, and red is the track of experimenter person-2.

As can be seen from Figure 9a, the positioning results of the UWB fluctuate wildly, and the UWB signals in corridors are often unstable. When positioning data of three or more base stations are not received at the same time, the positioning results cannot be solved, resulting in data loss in a certain period of pedestrian positioning. Figure 9b also shows that the visual positioning is relatively stable, with slight fluctuations and dense positioning points. However, due to the camera’s distortion of the picture, the error will gradually increase when the pedestrian target appears at the edge of the picture. Moreover, the experimenter person-2 was blocked by the pillars in the corridor for a period of time during the walking process. As a result, the human target cannot be accurately identified when it reappears in the field of vision; as a result, the tracker ID switches frequently and the positioning track fluctuates. In addition, the identity of the pedestrian based on the visual location alone is still unknown. However, it can be seen from Figure 9c that the results of the fusion positioning of the two experimenters are basically consistent with the real track, and their identities can be recognized. This is because the UWB positioning can compensate for the positioning results in the presence of the visual omission and blind areas, and the visual positioning can also supplement the positioning in the region with weak UWB signals. Furthermore, the stability of the visual positioning can effectively reduce the UWB positioning error, thus significantly improving the positioning results in fusion mode. The error diagram of pedestrian positioning in the *X*-axis direction is shown in Figure 10, and the error diagram in the *Y*-axis direction is shown in Figure 11. According to the error diagram, the error fluctuation of fusion positioning is smaller than that of the visual positioning and the UWB positioning. The error value is significantly lower than that of the visual positioning and the UWB positioning.

To better evaluate the effect of fusion positioning, the cumulative distribution function of the errors of the three positioning methods is plotted in Figure 12. It can be seen intuitively that, when CDF = 80%, the positioning accuracy of the UWB positioning method is about 39 cm, and the maximum error can be more than 60 cm. In the same way, the positioning accuracy of the visual positioning method is less than 36.2 cm. By comparison, 80% of the positioning errors of the fusion positioning method are within 24.3 cm, and the maximum error is 37 cm, which indicates its stronger robustness and less error fluctuation.

In order to further test the performance of this fusion algorithm, this paper designed experimental scenarios with different obstacles, placed as shown in Figure 13.

As shown in Figure 13, there are three chairs of different sizes and pillars as obstacles in the corridor. From Figure 14, it can be seen that when person-1 and person-2 are standing completely in front of different chairs, the visual detection tracking algorithm can still accurately identify the target in the tracking screen, and when person-3 is half blocked by the pillar, it is also able to identify the pedestrian target very well. It can be seen that visual positioning has high stability, and pedestrian targets can still be recognized and tracked when common obstacles appear in real life. The experimental results are shown in Figure 15.

In addition, we also investigated whether the length of time of the human target loss affects the accuracy of this method. The experimenters took the same route and included the blind area behind the column, but stayed behind the column for different periods of time; that is, the length of time of the human target loss was different. The experimental results are presented in Figure 16.

To better analyze the influence of different factors on the fusion positioning accuracy, we summarize the results in Table 1. The results show that the length of time of the human target loss does not have a significant impact on the accuracy because, within a certain time length threshold (3 s), the DeepSort algorithm has the ability to re-identify the target and will not reassign a new ID to the target. When the pedestrian target disappears beyond the threshold, although the DeepSort will reassign an ID number to the target, the UWB positioning results will be re-matched with the visual positioning results, and the accuracy of the fusion positioning results will not be affected after successful matching. In addition, we also found that, when placing ordinary obstacles in the environment, localization of the visual position does not result in a large amount of volatility. However, the presence of obstacles causes large fluctuations in the UWB localization results due to the NLOS errors in the UWB localization. Thus, it affects the results of fusion localization to a certain extent, but the fluctuations in the fusion localization results are significantly smaller than the results of localization by a single means.

### 5.2. Experimental Results of Kalman Filtering Based on the PCA

A Kalman filtering algorithm based on PCA is also proposed to improve the robustness of Kalman filtering. The experimental positioning results of different algorithms are shown in Figure 17. Starting from the beginning of the asterisk, the experimenter held the UWB tag and followed the red track. Then, the original positioning value of the UWB was filtered by the Kalman, and the blue dots in the figure show the positioning results processed by the PCA-Kalman filtering algorithm. It can be seen that the PCA-Kalman algorithm has a better filtering effect in the environment of colored noise, and the track is smoother and more consistent with the real track. This is due to the PCA preprocessing of the measured data. The colored noise, independent of the useful signal, is eliminated, and the Kalman gain is modified by degradation factors. The filtered trajectory is more consistent with the real scenario.

The error cumulative distribution functions of the three algorithms are shown in Figure 18. The positioning results of the proposed algorithm have an accuracy of 80% within 32 cm, and the mean square error is 31.8% lower than that of the conventional Kalman filter. In addition, the accuracy of the PCA-Kalman filter algorithm is better than the augmented Kalman filter algorithm in the experiment described in this paper. This is because the colored noise of the augmented Kalman filter algorithm needs to meet certain conditions and, as the dimension of the augmented system increases, the order of the filter also increases, resulting in a larger calculation. Experimental results prove that the proposed method can estimate the system state more effectively.

## 6. Conclusions

To meet the increasingly high demand for indoor positioning, in this paper, the advantages and disadvantages of a single UWB or visual positioning system are analyzed, and we illustrate the prominent positioning advantages of the fusion positioning system. On this basis, we presented an indoor multi-pedestrian target positioning method combining the UWB and vision to overcome the positioning defects of the single positioning system. In addition, to improve the positioning accuracy and reduce the adverse impact of colored noise on the state estimation of the fusion system, we proposed a Kalman filter algorithm based on PCA. The fusion positioning method of the UWB and vision can improve the problems of limited visual positioning range and missing trajectory, and effectively suppress the interference in the UWB positioning process. The experimental results show that the positioning performance of the fusion positioning system is better than that of the single positioning system, and the proposed PCA-Kalman filtering algorithm is superior to the conventional filtering algorithm in terms of positioning accuracy and robustness. The PCA algorithm is used to preprocess the measured data and reduce the cross-correlation of data samples to its minimum. Consequently, the colored noise is reduced and the Kalman gain is improved. The experimental results confirmed the new filtering algorithm has better performance in a colored noisy environment.

In future research, it is expected that the above fusion systems will be applied to seamless indoor and outdoor positioning. Another area for future research would be exploring the PCA-Kalman filtering in more complex scenarios, such as when the pedestrian trajectory is curved.

## Figures and Tables

**Figure 1 sensors-22-01394-f001:**
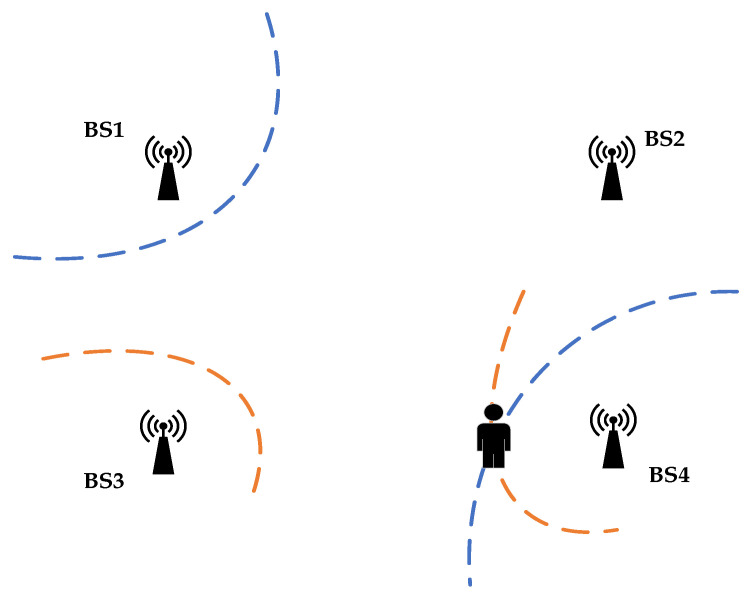
TDOA positioning schematic diagram.

**Figure 2 sensors-22-01394-f002:**
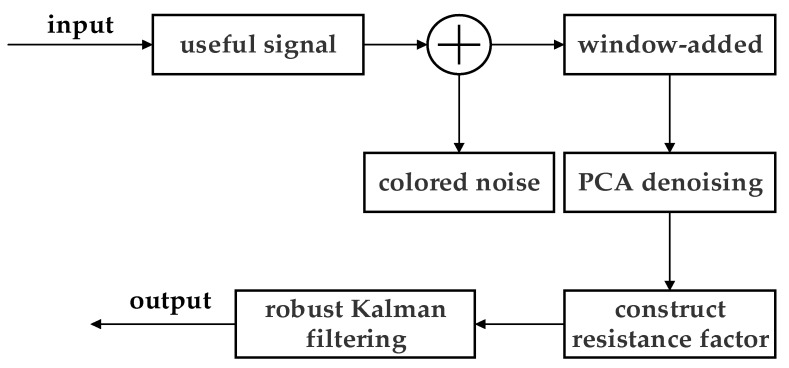
PCA–Kalman algorithm block diagram.

**Figure 3 sensors-22-01394-f003:**
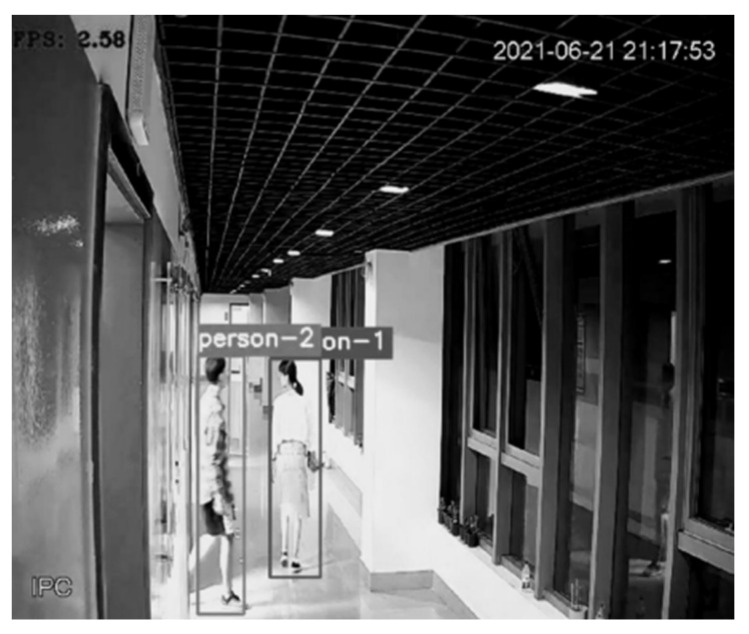
Visual target tracking results.

**Figure 4 sensors-22-01394-f004:**
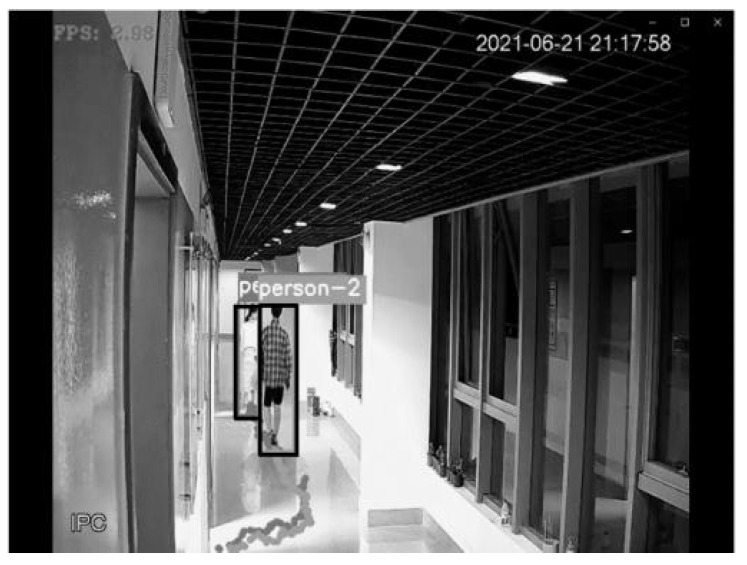
Results of visual persistent positioning.

**Figure 5 sensors-22-01394-f005:**
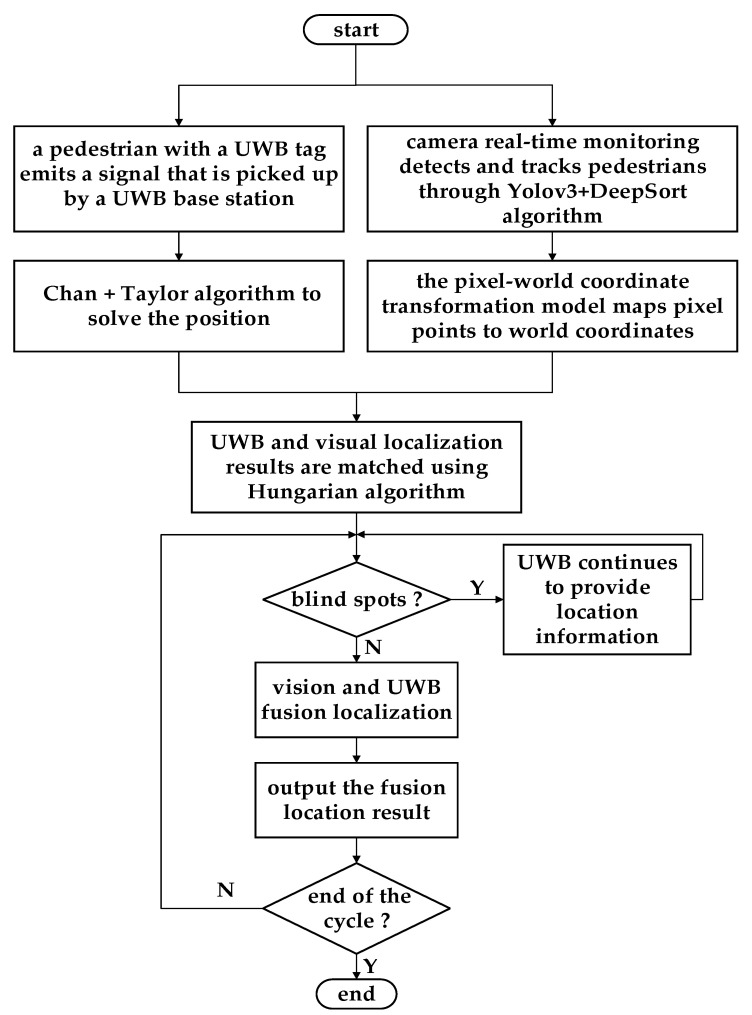
Flowchart of the fusion location algorithm model.

**Figure 6 sensors-22-01394-f006:**
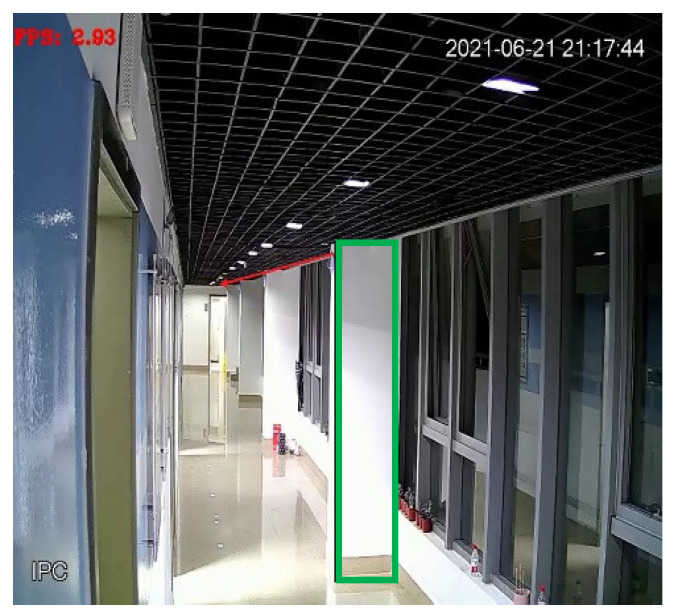
Experimental Environment.

**Figure 7 sensors-22-01394-f007:**
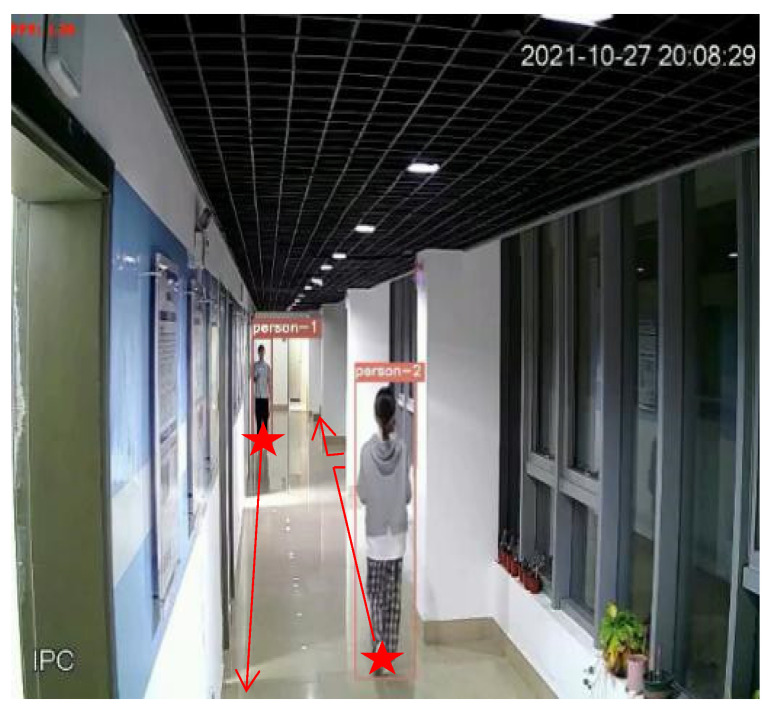
Experimental Scenario 1.

**Figure 8 sensors-22-01394-f008:**
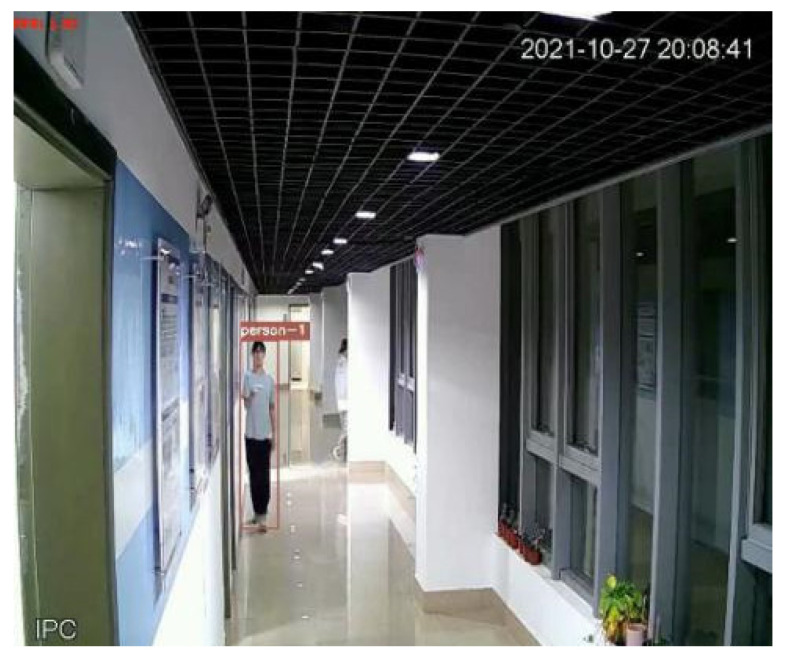
Experimental Scenario 2.

**Figure 9 sensors-22-01394-f009:**
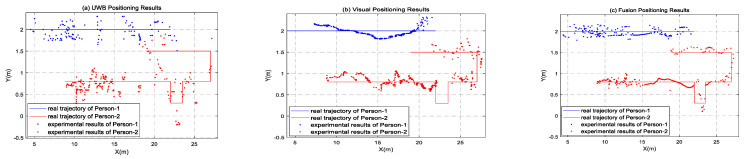
Positioning results.

**Figure 10 sensors-22-01394-f010:**
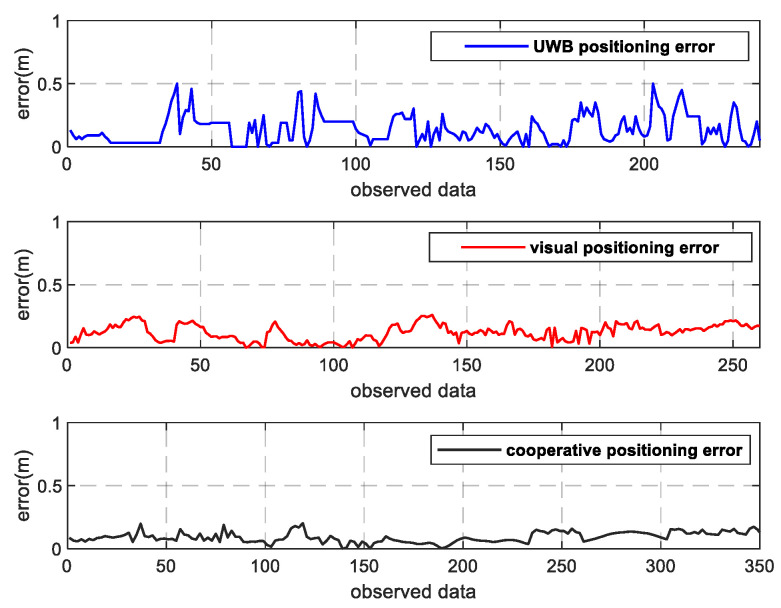
Error of the *X*-axis.

**Figure 11 sensors-22-01394-f011:**
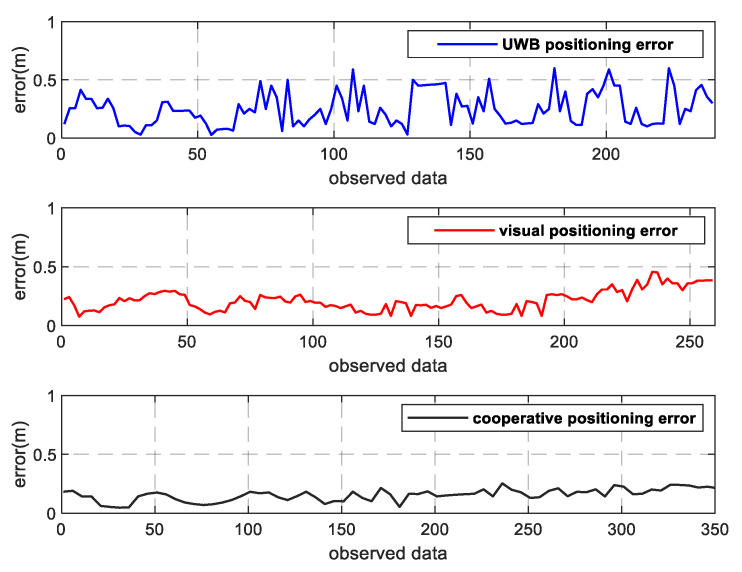
Error of the *Y*-axis.

**Figure 12 sensors-22-01394-f012:**
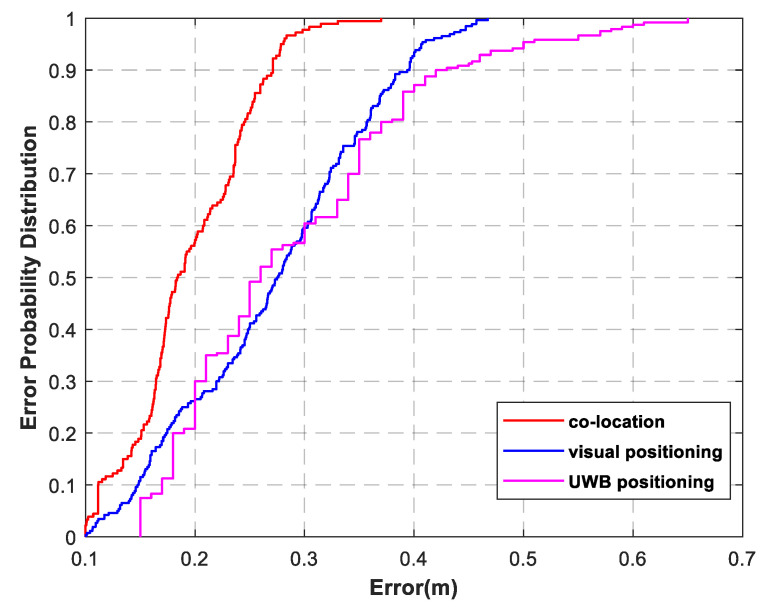
Error probability distribution.

**Figure 13 sensors-22-01394-f013:**
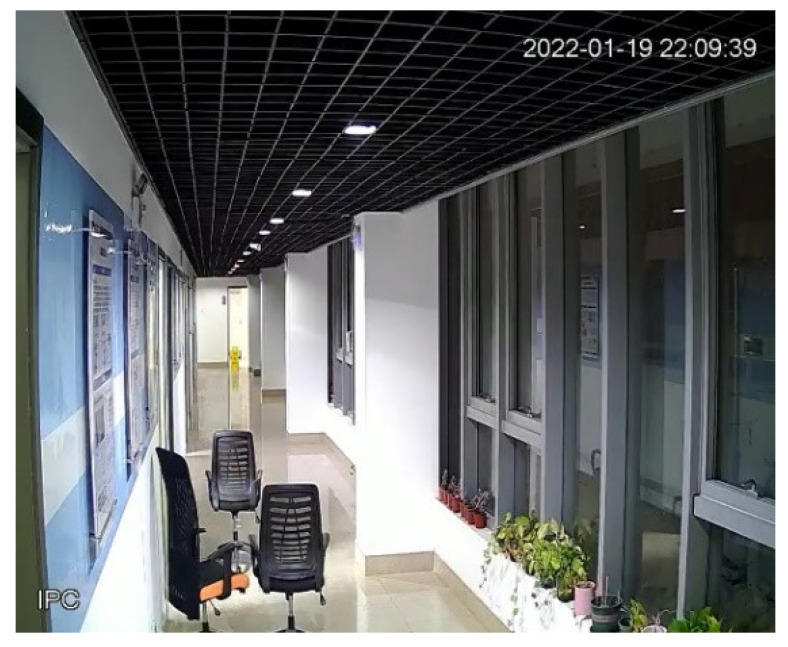
Experimental scenario with three obstacles.

**Figure 14 sensors-22-01394-f014:**
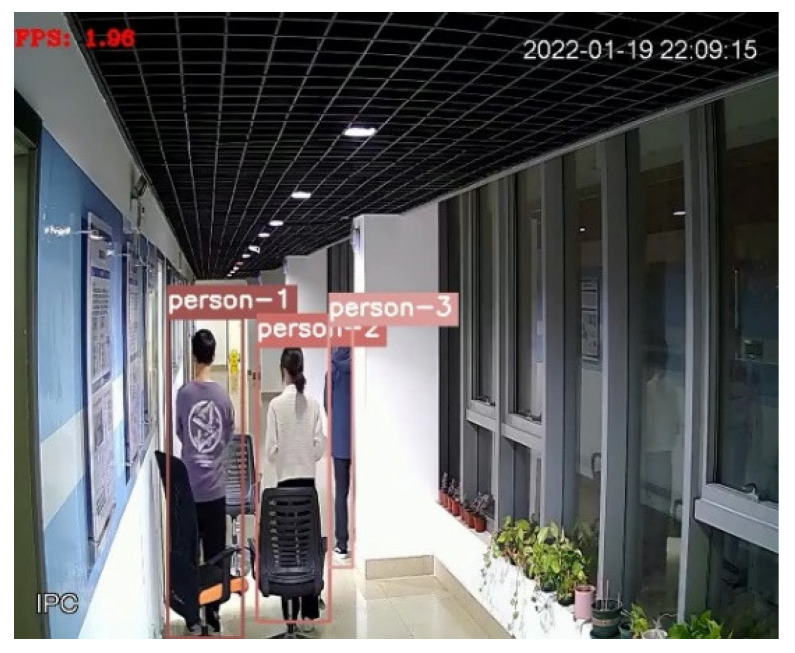
The effect of the DeepSort algorithm with obstacles.

**Figure 15 sensors-22-01394-f015:**
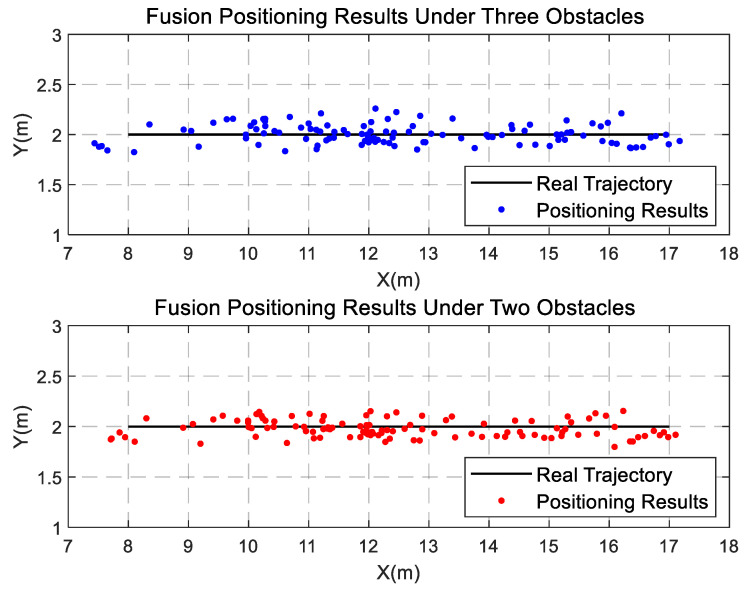
Fusion positioning results with different number of obstacles.

**Figure 16 sensors-22-01394-f016:**
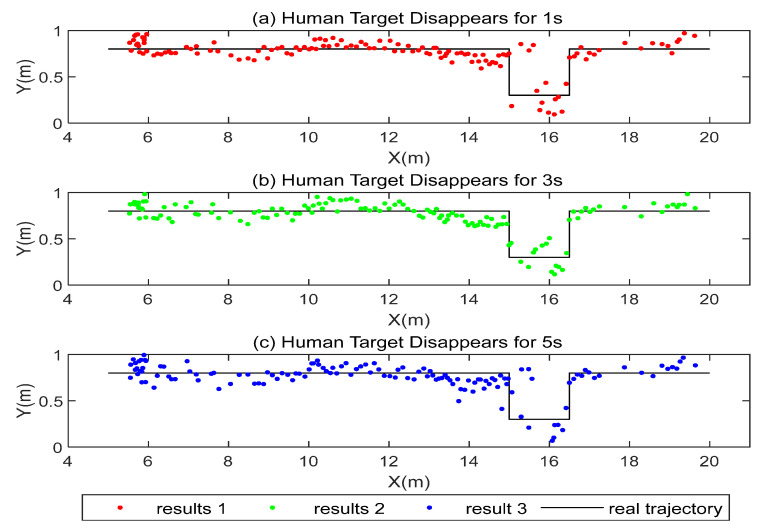
Fusion positioning results with different lengths of time of the human target loss.

**Figure 17 sensors-22-01394-f017:**
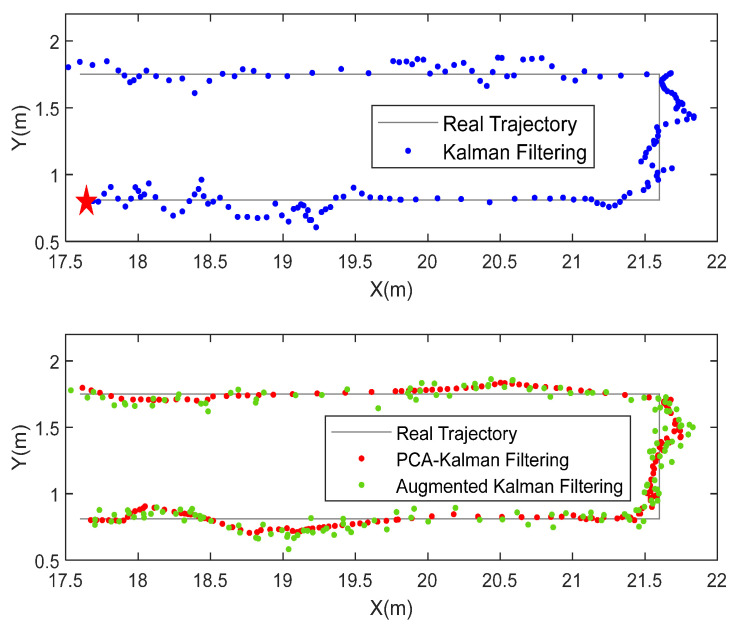
Positioning results. The experimenter starts from the red star.

**Figure 18 sensors-22-01394-f018:**
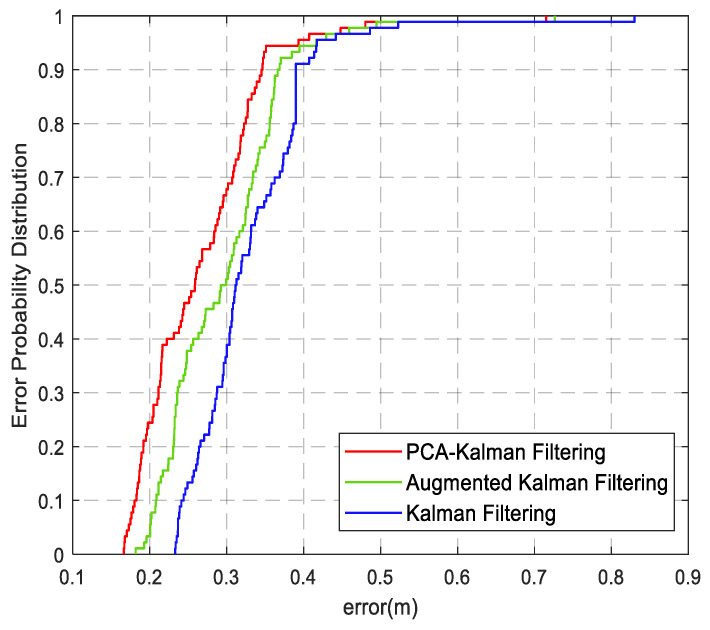
Error probability distribution.

**Table 1 sensors-22-01394-t001:** Positioning results under different experimental conditions.

Scheme	Number of Human Targets	Time Length of Human Target Loss/s	Number of Obstacles	Mean Error of Visual Positioning/cm	Mean Error of UWB Positioning/cm	Mean Error of Fusion Positioning/cm
①	2	3	0	30.16	33.25	22.43
②	1	3	0	29.72	30.89.	20.94
③	1	5	0	29.45	32.23	21.28
④	1	1	0	28.58	33.99	21.99
⑤	1	0	2	27.73	36.58	23.45
⑥	1	0	3	28.25	38.56	25.08

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
