# Peer review of "An Indoor Positioning Method Based on UWB and Visual Fusion"

_sensors, 2022, doi:10.3390/s22041394_

Round 1

Reviewer 1 Report

I would be very glad to review the manuscript in greater depth once it has been edited because the subject is interesting. This manuscript proposed a novel indoor positioning method fusing ultra-Wide Band (UWB) and vision, and applied a Kalman filter algorithm based on PCA to improve positioning accuracy. However, I thought it still has some deficiencies and I recommend to a major revision before acceptable publication. Detailed comments are listed below: 

--Section Abstract: The author needs to streamline the content of Abstract. The authors need to emphasis the innovation of the paper, and what problem the proposed method has solved while other methods cannot solve.
--Section 1: The authors mention “there is lots of colored noise in indoor positioning observables, and a single positioning system cannot meet the requirement for positioning accuracy.” in this section. The authors need to explain whether all positioning system are affected by colored noise.
--Section 1: In the first paragraph of the section introduction, the authors firstly highlight the negative impact of colored noise, and then mention “the research and development of accurate and reliable indoor positioning methods can not only improve the accuracy and reliability of indoor target positioning, but also have a broad market application prospect”. The author needs to explain whether only the denoise method is applied to improve the position accuracy. The authors need to emphasis the significance of a novel indoor positioning method fusing UWB and vision.
--Section 1: A more detailed description of previous research in introduction is needed, especially the published literatures on location for other positioning methods, for example: Velocity-Free Localization of Autonomous Driverless Vehicles in Underground Intelligent Mines IEEE Transactions on Vehicular Technology, 2020, 69(9), 9292 – 9303; Enhanced Autonomous Exploration and Mapping of an Unknown Environment with the Fusion of Dual RGB-D Sensors,ENGINEERING, 2019, 5(1), 164-172; Hybrid ToA and IMU indoor localization system by various algorithms JOURNAL OF CENTRAL SOUTH UNIVERSITY, 2019, 26(8), 2281-2294.
--Section 2: The authors mention “Kalman filter algorithm has excellent filtering effect when both dynamic noise and measurement noise are white noise. However, since most of the actual engineering is colored noise, the effectiveness and stability of the filter will be reduced” in section 2.3. The authors need to specify what the difference is between white noise and colored noise and how the colored noise affect the Kalman filter algorithm.
--Section 5: It is recommended to complement the labels of each subplot in Figure 9, which will help readers understand. Besides, it is suggested to improve the quality of Figure 9 in a more stereo and intuitive way.
--Section 5: The proposed fusion positioning takes advantages of UWB and visual positioning. It is suggested to investigate whether the time length of human target loss affects the accuracy of this method.
--Section 5: It is recommended to complement content of the experiment part. Whether factors such as the number of targets, the route, the size of the mask and its number have an impact on the accuracy of the proposed positioning algorithm?
--Please refer to the format of the journal to improve the manuscript.
--Tables: Please refer to the document to modify the format of the tables.

Reviewer 2 Report

The paper is well written and seems solid in terms of its fundamental ideas. 

Nevertheless, a proofread by a native English speaker would be gainful.

The experimental results outperforms other approaches.

Please develop more future work details in the last section. You can add a future work subsection for that.

Also, you can present a concrete example to demonstrate how this system could be used in the real-world.

All in all, this study can bridge the gap between theory and practical implementation, providing us with a solution in a decent amount of time. Based on that, I recommend this paper for publication.

Reviewer 3 Report

1 The introduction provide insufficient background and references is obsolete.

2 There are some grammatical errors in the paper, especially in the usage of the crown

3 The contribution seems limited.

4 Figure presentation has some problem,for example: Figure 13.

Reviewer 4 Report

The authors present a uwb/visual fusion method for pedestrian tracking. Regarding this study i have following points and i hope the authors can address them.

  1. How do you define good signal and noise signal in pca? What is the definition of this noise. E.g. if there is a very long obstacle and the uwb signal is always on nlos path, the resulted target position is always biased. How can pca capture such bias or the cause of bias?
  2.  How do you select R and Q for Kalman filter? 
  3. Another problem for this study is that the introduction of each method is too textbookish. It would be better to include some practice example to make the reader understand the concept or idea of the paper easily. E.g an example of good signal and noise signal of pca.
  4. Last point, i am not very clear about Kalman filter part. Is it used for uwb tracking or fusion as well? Please describe Kalman in more detail, e.g. what are estimated in state. I have a feeling that the first part of paper gives more detail but second part is not very clear. 
  1.  

Round 2

Reviewer 1 Report

The manuscript did not revise per first round comments.

The authors did not revise well for all comments from reviewers and editor, such as for comments:  Point 1: The author needs to streamline the content of Abstract. The authors need to emphasis the innovation of the paper, and what problem the proposed method has solved while other methods cannot solve.

The authors used more than 60% words to introduce the background,but did not highlight their method in a detail and necessary way. Furthermore, the abstract only has some results without basic conclusions.

Reviewer 3 Report

This paper named "An Indoor Positioning Method Based on UWB and Visual Fusion" proposed an indoor multi-pedestrian target positioning method combining UWB and vision to overcome the positioning defects of the single positioning system.  The experimental results have confirmed the new filtering algorithm has better performance in a colored noisy environment.  In addition, this paper describes a new method for preparation of the PCA-Kalman filtering algorithm. The text is well arranged and the experimental logic is clear. However, there also are some English mistakes in the manuscript. For example:

(1)Taylor algorithm is a recursive algorithm  --> The Taylor algorithm is a recursive algorithm

 (2),otherwise The Taylor algorithm... -->  ,otherwise the Taylor algorithm...

 (3)...

Please check again carefully!

Moreover, figure (1), (2) and (5) should be as the same clearly as figure (9) or (15).

Reviewer 4 Report

The authors have addressed all my points. The paper is ready for publishing. 
